# SELF-SUPERVISED DEBIASING USING LOW RANK REGULARIZATION

## ABSTRACT

Spurious correlations can cause strong biases in deep neural networks, impairing generalization ability. While most existing debiasing methods require full supervision on either spurious attributes or target labels, training a debiased model from a limited amount of both annotations is still an open question. To address this issue, we investigate an interesting phenomenon using the spectral analysis of latent representations: spuriously correlated attributes make neural networks inductively biased towards encoding lower effective rank representations. We also show that a rank regularization can amplify this bias in a way that encourages highly correlated features. Leveraging these findings, we propose a self-supervised debiasing framework potentially compatible with unlabeled samples. Specifically, we first pretrain a biased encoder in a self-supervised manner with the rank regularization, serving as a semantic bottleneck to enforce the encoder to learn the spuriously correlated attributes. This biased encoder is then used to discover and upweight bias-conflicting samples in a downstream task, serving as a boosting to effectively debias the main model. Remarkably, the proposed debiasing framework significantly improves the generalization performance of self-supervised learning baselines and, in some cases, even outperforms state-of-the-art supervised debiasing approaches.

## 1 INTRODUCTION

While modern deep learning solves several challenging tasks successfully, a series of recent works (Geirhos et al., 2018; Gururangan et al., 2018; Feldman et al., 2015) have reported that the high accuracy of deep networks on in-distribution samples does not always guarantee low test error on out-of-distribution (OOD) samples, especially in the context of spurious correlations. Existing studies (Arjovsky et al., 2019; Nagarajan et al., 2020; Tsipras et al., 2018) suggest that the deep networks can be potentially biased to the spuriously correlated attributes, or dataset bias, which are misleading statistical heuristics that are closely correlated but not causally related to the target label.

These catastrophic pitfalls of dataset bias have facilitated the development of debiasing methods, which can be roughly categorized into approaches: (**1**) leveraging annotations of spurious attributes, i.e., bias label (Kim et al., 2019; Sagawa et al., 2019; Wang et al., 2020; Tartaglione et al., 2021); (**2**) presuming specific type of bias, e.g., color and texture (Bahng et al., 2020; Wang et al., 2019; Ge et al., 2021); or (**3**) without using explicit kinds of supervisions on dataset bias (Liu et al., 2021; Nam et al., 2020; Lee et al., 2021; Levy et al., 2020; Zhang et al., 2022).

While substantial advances have been made in this regard, these approaches still fail to address the problem: how to train a debiased classifier by fully exploiting unlabeled samples lacking *both* bias and target label. More specifically, while the large-scale unlabeled dataset can be potentially biased towards spuriously correlated sensitive attributes, e.g., ethnicity, gender, or age (Abid et al., 2021; Agarwal et al., 2021), current existing debiasing frameworks are not designed to deal with this real-world unsupervised settings. Here we also confirm that most supervised debiasing frameworks suffer from performance degradation in the low-labeled data setting. Moreover, recent works have suggested that self-supervised learning might not be sufficient to deal with OOD generalization (Geirhos et al., 2020; Chen et al., 2021; Robinson et al., 2021) when dataset bias remains after data augmentation.

To tackle this issue, we first make a series of empirical observations that allow us to examine the fundamental difference between biased and unbiased representations. Interestingly, we found that

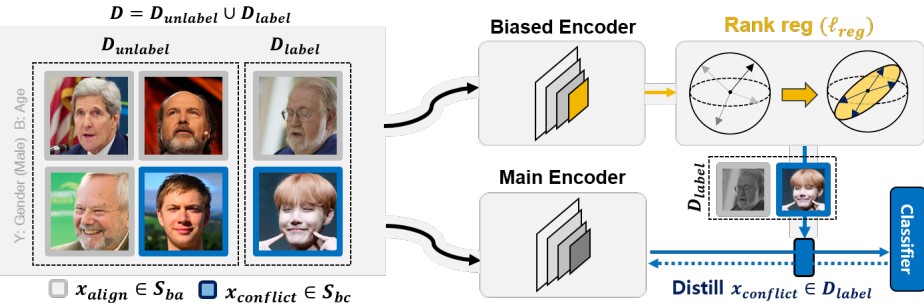

Figure 1: **Concept.** Based on the bias-rank relationship (Section 2), we introduce a novel debiasing framework centered on rank regularization, which intentionally amplifies spurious correlation by enforcing feature components to be *entangled* with both spurious and invariant attributes.

spurious correlations suppress the effective rank (Roy & Vetterli, 2007) of latent representations, which severely deteriorates the semantic diversity of representations and leads to the degradation of feature discriminability. Another notable aspect of our findings is that the intentional increase of feature redundancy amplifies "prejudice" in neural networks. To be specific, as we enforce the correlation among latent features to regularize the effective rank of representations (i.e., rank regularization), the accuracy on bias-conflicting samples quickly declines while the model still performs reasonably well on the bias-aligned [1] samples.

Based on these observations, we propose a novel debiasing framework that can utilize both labeled *and* unlabeled biased samples with rank regularization. The proposed method is fully compatible with both supervised and self-supervised scenarios, where such compatibility arises from the rank regularization that does *not* rely on any labels. Specifically, for a supervised (self-supervised) setting, we train 1) a biased classifier (encoder) with rank regularization, which serves as a semantic bottleneck limiting the semantic diversity of feature components, and 2) the main classifier (encoder) with standard (self-)supervised learning approaches. The biased model affords us the leverage to uncover spurious correlations and identify bias-conflicting samples in a downstream task.

Our work is the first to unveil the bias-rank relationships and introduce an effective debiasing strategy to exploit potentially unlabeled data samples. We demonstrate the effectiveness of the proposed debiasing framework with various challenging real-world biased datasets, including MultiCMNIST (Li et al., 2022), biased Chest X-ray databases, UTKFace, CelebA, etc., in both a supervised and self-supervised scenario. These experiments show that our method significantly outperforms other self-supervised baselines, and even state-of-the-art supervised debiasing methods in some cases.

## 2 LOW-RANK BIAS OF BIASED REPRESENTATIONS

### 2.1 PRELIMINARIES

Throughout the paper, we denote $x \in \mathbb{R}^m$ and $y \in \mathcal{Y}$ as $m$-dimensional input sample and its corresponding predicting label, respectively. Then we denote $X = \{x_k\}_{k=1}^n$ as a batch of $n$ samples from a dataset which is fed to an encoder $f_\theta : \mathbb{R}^m \to \mathbb{R}^d$, parameterized by $\theta$. Then we construct a matrix $Z \in \mathbb{R}^{n \times d}$ where each $i$th row is the output representations of the encoder $f_\theta(x_i)^T$ for $x_i \in X$. For every analysis in this section, we use $Z$ as our latent representations, where the neural backbone of the encoder may vary as simple convolutional networks, ResNet-18, or ViT (Dosovitskiy et al., 2020) (Experimental details provided in Appendix C.1 and D).

To evaluate the semantic diversity of given representation matrix, we introduce *effective rank* (Roy & Vetterli, 2007) which is a widely used metric to measure the effective dimensionality of matrix and analyze the spectral properties of features in neural networks (Arora et al., 2019; Razin & Cohen, 2020; Huh et al., 2021; Baratin et al., 2021):

---

[1]The *bias-aligned* samples refer to data with a strong correlation between (potentially latent) spurious features and target labels. The *bias-conflicting* samples refer to the opposite cases where spurious correlations do not exist.

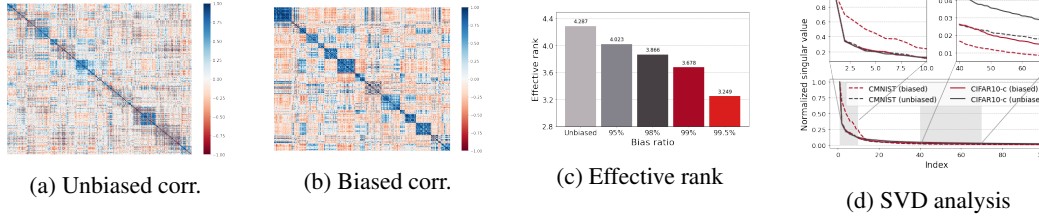

(a) Unbiased corr.   (b) Biased corr.   (c) Effective rank

(d) SVD analysis

Figure 2: Empirical analysis on rank reduction phenomenon. For every analysis, we used the output $Z$ of the encoder (Sec. 2.1). (**a**, **b**): Hierarchically clustered auto-correlation matrix of unbiased and biased representations (Bias ratio=99%). (**c**): Effective rank with color bias. 'Unbiased' represents the case training model with perfectly unbiased dataset, i.e. random color for each sample. (**d**): SVD analysis with max-normalized singular values. Top 100 values are shown in the figure (total: 256).

**Definition 2.1.** Given the matrix $X \in \mathbb{R}^{m \times n}$ and its singular values $\{\sigma_i\}_{i=1}^{\min(m,n)}$, the effective rank $\rho$ of $X$ is defined as the shannon entropy of normalized singular values:

$$\rho(X) = - \sum_{i=1}^{\min(m,n)} \bar{\sigma}_i \log \bar{\sigma}_i, \qquad (1)$$

where $\bar{\sigma}_i = \sigma_i / \sum_k \sigma_k$ is $i$-th normalized singular value. Without loss of generality, we omit the exponentiation of $\rho(X)$ as done in (Roy & Vetterli, 2007).

Effective rank is also referred to as spectral entropy where its value is maximized when the singular values are all equal and minimized when a top singular value dominates relative to all others. Recent works (Chen et al., 2019b;a) have revealed that the discriminability of representations resides on wide range of eigenvectors since the rich discriminative information for the classification task cannot be transmitted by only few eigenvectors with top singular values. Thus from a spectral analysis perspective, effective rank quantifies how diverse the semantic information encoded by each eigenfeature is, which is closely related to the feature discriminability across target label categories. In the rest of paper, we interchangeably use effective rank and rank by following prior works.

## 2.2 SPECTRAL ANALYSIS OF THE BIAS-RANK RELATIONSHIPS

We now present experiments showing that the deep networks may tend to encode lower-rank representations in the presence of stronger spurious correlations. To arbitrarily control the degree of spurious correlations, we introduce synthetic biased datasets, Color-MNIST (CMNIST) and Corrupted CIFAR-10 (CIFAR-10C, (Hendrycks & Dietterich, 2019)), with color and corruption bias types, respectively. We define the degree of spurious correlations as the ratio of bias-aligned samples included in the training set, or bias ratio, where most of the samples are bias-aligned in the context of strong spurious correlations.

Figure 2c shows that the rank of latent representations from a penultimate layer of the simple convolutional classifier decreases as the bias ratio increases in CMNIST. We provide similar rank reduction results of CIFAR-10C with ResNet-18 and ViT in the Appendix C.1. We further compare the correlation matrix of biased and unbiased latent representations in the penultimate layer of biased and unbiased classifiers, respectively. In Figure 2a and 2b, we observe that the block structure in the correlation matrix is more evident in the biased representations after the hierarchical clustering, indicating that the features become highly correlated which may limit the semantic diversity of networks. To investigate the rank reduction phenomenon in-depth, we compare the normalized singular values of biased and unbiased representations. We conduct singular value decomposition (SVD) on the feature matrices of both biased and unbiased classifiers and plot the singular values normalized by the spectral norm of the corresponding matrix. Figure 2d shows that the top few normalized singular values of biased representations are similar to or even greater than those of unbiased representations. However, the remaining majority of singular values decay significantly faster in biased representations, greatly weakening the informative signals of eigenvectors with smaller singular values and deteriorating feature discriminability (Chen et al., 2019b;a).

## 2.3 RANK REGULARIZATION

Motivated from the aforementioned rank reduction phenomenon, we ask an opposite-directional question: "Can we intentionally amplify the prejudice of deep networks by *maximizing* the redundancy

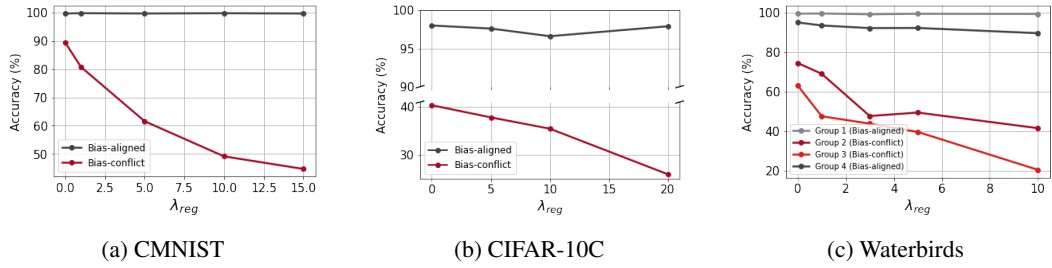

(a) CMNIST          (b) CIFAR-10C          (c) Waterbirds

Figure 3: (**a**, **b**): Bias-conflict and Bias-aligned accuracy on CMNIST and CIFAR-10C (Bias ratio=95%). (**c**): Group accuracy on Waterbirds. Detailed simulation settings are in the Appendix D.

|  | CMNIST | | CIFAR-10C | |
|---|---|---|---|---|
|  | P (%) | R (%) | P (%) | R (%) |
| ERM | 85.59 | 19.76 | 52.03 | 0.06 |
| + Rank reg | **98.83** | **95.91** | **71.39** | **51.43** |

(a) CMNIST, CIFAR-10C

| Metrics | ERM | JTT | Rank reg |
|---|---|---|---|
| Precision (%) | 37.84 | 48.95 | **54.77** |
| Recall (%) | 11.67 | 48.75 | **55.01** |

(b) Waterbirds

Table 1: Precision (P) and Recall (R) of bias-conflicting samples. (**a**): Bias-conflicting samples are identified in the error set of ERM model trained with and without rank regularization (Bias ratio=95% for both datasets). (**b**): Bias-conflicting samples are similarly identified by ERM, JTT, and the proposed biased model in Waterbirds dataset.

between the components of latent representations?". If the feature components are extremely correlated, the corresponding representations may exhibit most of its spectral energy along the direction of one singular vector. For this case, effective rank may converge to 0. In other words, our goal is to design a *semantic bottleneck* of representations that restricts the semantic diversity of feature vectors. To implement the bottleneck in practice, motivated from Figure 2b, we compute the auto-correlation matrix of the output of encoder.

Let $\bar{Z}$ denote the mean-centered representations $Z$ along the batch dimension. The normalized auto-correlation matrix $C \in \mathbb{R}^{d \times d}$ of $\bar{Z}$ is defined as follow:

$$C_{i,j} = \frac{\sum_{b=1}^{n} \bar{Z}_{b,i}\bar{Z}_{b,j}}{\sqrt{\sum_{b=1}^{n} \bar{Z}_{b,i}^2}\sqrt{\sum_{b=1}^{n} \bar{Z}_{b,j}^2}} \quad 1 \leq \forall i, j \leq d, \tag{2}$$

where $b$ is an index of sample and $i, j$ are index of each vector dimension. Then we define our regularization term as the negative of a sum of squared off-diagonal terms in $C$:

$$\ell_{reg}(X; \theta) = -\sum_{i}\sum_{j \neq i} C_{i,j}^2, \tag{3}$$

where we refer to it as the *rank loss*. Note that the target labels on $X$ is *not* used at all.

**Analysis of rank-regularized networks.** To investigate the impacts of rank regularization in deep neural networks, we construct the classification model by combining the linear classifier $f_W : \mathbb{R}^d \to \mathbb{R}^c$ parameterized by $W \in \mathcal{W}$ on top of the encoder $f_\theta$, where $c = |\mathcal{Y}|$ is the number of classes. Then we trained models by cross entropy loss $\ell_{CE}$ combined with $\lambda_{reg}\ell_{reg}$, where $\lambda_{reg} > 0$ is a Lagrangian multiplier. We use CMNIST, CIFAR-10C, and Waterbirds dataset (Wah et al., 2011), and evaluate the trained models on an unbiased test set following Nam et al. (2020); Lee et al. (2021). After training models with varying the hyperparameter $\lambda_{reg}$, we compare bias-aligned and bias-conflict accuracy, which are the average accuracy on bias-aligned and bias-conflicting samples in the unbiased test set, respectively, for CMNIST and CIFAR-10C. Test accuracy on every individual data group is reported for Waterbirds. Figure 3 shows that models suffer more from poor OOD generalization as trained with larger $\lambda_{reg}$. The average accuracy on bias-conflicting groups is significantly degraded, while the accuracy on bias-aligned groups is maintained to some extent. It implies that rank regularization may force deep networks to focus on spurious attributes.

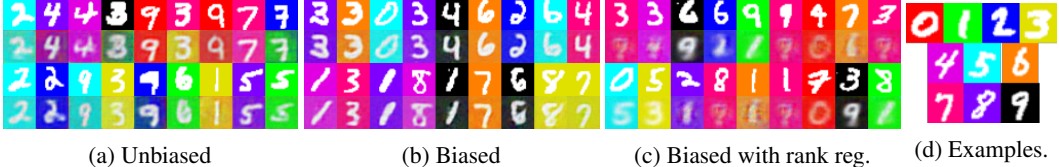

| (a) Unbiased | (b) Biased | (c) Biased with rank reg. | (d) Examples. |

Figure 4: Randomly selected reconstructed images from representations with varying degrees of bias. First and third row correspond to the input bias-conflicting images. Second and fourth row correspond to the reconstructed images. Reconstructed from (**a**) unbiased representations, (**b**) biased representations, and (**c**) biased representations with rank regularization (bias ratio=95% in **b, c**). (**d**) Examples of bias-aligned CMNIST images.

**Minority mining performance.** Table 1a and 1b support that the biased models with strong regularization can effectively probe out the bias-conflicting samples in the training set. Specifically, we train a biased classifier with rank regularization and distill an error set $E$ of misclassified training samples as bias-conflicting samples proxies. As reported in Table 1a, we observe that our biased classifier is relatively robust to the unintended memorization of bias-conflicting samples (Sagawa et al., 2020) in contrast to the standard models trained by Empirical Risk Minimization (ERM). Moreover, Table 1b shows that the proposed rank regularization improves the precision and recall of identified bias-conflicting samples compared to JTT (Liu et al., 2021). Detailed simulation settings are in the Appendix D.

**Reconstruction of biased representations.** To understand the relationship between rank regularization and spurious correlations more deeply, we visualize the pretrained representations with varying degrees of bias. We first trained deep networks on: (**a**) unbiased CMNIST (random background color), (**b**) biased CMNIST (bias ratio=95%) without rank regularization and (**c**) with rank regularization ($\lambda_{reg} = 50$). Then, we train the auxiliary decoder, which reconstructs the bias-conflicting images from the freezed latent representations of each pretrained network. Results show that rank regularization may cause the representation to lose information on complex invariant features, resulting in a loss of feature discriminability and informative signals. While both digit and color are well reconstructed with biased representations (**b**), the decoder fails to reconstruct bias-conflicting images from the (**c**) biased representations pretrained with rank regularization. The foreground digit is blurred, and its class changes following the color-digit assignment in Figure 4d.

These observations afford us some key insights into rank regularization: First, the rank-regularized representation may lose its information on complex invariant features (i.e., shape and style of the foreground digit), specifically undermining the feature discriminability and informative signals. Second, the limited semantic diversity makes it harder to identify the true underlying independent generative factors for multidimensional data; instead, it may encode feature components *entangled* with both spurious and invariant attributes as the digit class of the reconstructed image is erroneously determined by the background color in 4c.

**Multiple bias attributes.** To further investigate the generalizability of rank regularization, we evaluate the biased representations with Multi-Color MNIST (MultiCMNIST) dataset (Li et al., 2022), which is similar to the CMNIST but have two bias attributes: left and right background colors. We set bias ratio=99% for the left color and bias ratio=95% for the right color, i.e., the left color is a more salient bias than the right color (Dataset details are provided in Appendix D).

Table 2 shows that the rank regularization successfully biases the model w.r.t both bias attributes, while LfF (Nam et al., 2020) completely fails to amplify the right color bias, i.e. less salient bias, as shown in the second row (Biased accuracy part). This leads to the abnormal debiasing results of LfF as shown in Table 2 where it records unbalanced accuracy for the left- and right-color-bias-conflicting samples. In contrast, the proposed framework shows superior performance by simply upweighting the misclassified bias-conflicting proxies, as done in Liu et al. (2021).

Taken together, these results indicate that the rank regularization encourages the network to focus more on spurious correlations in a way that minimizes semantic diversity and *entangles* invariant and spurious features (Park et al., 2023), which is a fundamentally different mechanism compared to the LfF (Nam et al., 2020) with its easy-to-learn assumption. More details on the upweighting strategy will be provided in Section 3 and pseudo-code in Appendix A.

Table 2: (**a**) Test accuracy (%) on MultiCMNIST. Lower is better for this results. BC for bias-conflicting, and BA for bias-aligned. Bias ratio=99(%) for left color, and 95(%) for right color. $\lambda_{reg} = 50$ is used for rank regularization. (**b**) Debiasing results. Higher is better for this results. Baseline results are from Li et al. (2022). $\lambda_{up} = 50$ is used for upweighting in the proposed framework ($\lambda_{up}$: a manual rescaling weight given to each identified bias-conflicting samples in cross entropy loss). Pseudo-code and experimental details are provided in Appendix A and D, respectively.

| Idx | Left color | Right color | (a) Biased accuracy (%) | | | (b) Debiased accuracy (%) | | |
| --- | --- | --- | --- | --- | --- | --- | --- | --- |
| | | | ERM | LfF (Nam et al., 2020) | **Rank reg.** | LfF | DebiAN (Li et al., 2022) | **Ours** |
| (1) | BA | BC | 100.0 | 100.0 | 100.0 | 99.6 | 100.0 | 100.0 |
| (2) | BA | BC | 96.6 | 98.8 | 41.6 | 4.7 | 95.6 | 97.0 |
| (3) | BC | BA | 29.3 | 3.2 | 8.7 | 98.6 | 76.5 | 79.1 |
| (4) | BC | BC | 7.6 | 1.3 | 6.1 | 5.1 | 16.0 | 18.3 |
| (1) $\sim$ (4) average acc. | | | 58.38 | 50.83 | **39.1** | 52.0 | 72.0 | **73.6** |

# 3 DeFund: Debiasing framework with unlabeled data

Motivated by the observations in Section 2, we propose a self-supervised debiasing framework with unlabeled data, coined DeFund (Debiasing Framework with Unlabeled Data). A notable distinction from previous studies (Bahng et al., 2020; Zhang et al., 2022) lies in the proposed framework's ability to effectively harness unlabeled data for learning biased representations. This is achieved through the application of self-supervised learning and rank regularization techniques.

The proposed framework is composed of two stages: We first train the biased encoder, which can be potentially adopted to detect the bias-conflicting samples in a downstream task, along with the main encoder by self-supervised learning, both without any labels. After pretraining, we identify the bias-conflicting samples in the downstream task using linear evaluation protocol (Oord et al., 2018; Chen et al., 2020). This set of samples serves as a boosting to debias the main model.

**Notation.** We denote $f_\theta^{bias} : \mathcal{X} \to \mathbb{R}^d$ and $f_\phi^{main} : \mathcal{X} \to \mathbb{R}^d$ as biased encoder and main encoder parameterzied by $\theta \in \Theta$ and $\phi \in \Theta$, respectively, where $d$ is the dimensionality of latent representations. Then we can compute the rank loss in (3) with introduced encoders and given batch $\{x_k\}_{k=1}^N$ with size $N$. Let $f_{W_b}^{cls} : \mathbb{R}^d \to \mathbb{R}^C$ be a single-layer classifier parameterized by $W_b \in \mathcal{W}$ which is placed on top of biased encoder $f_\theta^{bias}$, where $C = |\mathcal{Y}|$ is the number of classes. We similarly define the linear classifier $f_{W_m}^{cls}$ for the main encoder. Then we refer to $f^{bias} : \mathcal{X} \to \mathbb{R}^C$ as biased model, where $f^{bias}(x) = f_{W_b}^{cls}(f_\theta^{bias}(x)), \forall x \in \mathcal{X}$. We similarly define the main model $f^{main}$ as $f^{main}(x) = f_{W_m}^{cls}(f_\phi^{main}(x)), \forall x \in \mathcal{X}$. While the projection networks (Chen et al., 2020) are employed as well, we omit the notations because they are not engaged in classification.

**Stage 1. Training a biased encoder.** To train the biased encoder $f_\theta^{bias}$, we revisit the proposed rank regularization term (3) in context of instance discrimination task. Building upon the observations in Section 2.3, we conjecture that rank regularization may amplify bias in self-supervised learning as well by entangling invariant and spurious features. Based on these intuitions, we apply rank regularization directly to the output of the base encoder, which encourages each feature component to be highly correlated. From these applications, several noteworthy observations have emerged: (**a**) The representation becomes more biased as it is trained with stronger regularization (Appendix C.1). (**b**) While the overall performance may be upper-bounded due to the constraint on effective dimensionality (Jing et al., 2021), the bias-conflict accuracy is primarily sacrificed compared to the bias-aligned accuracy (Section 4).

**Stage 2. Debiasing downstream tasks.** After training the biased encoder, our goal is to debias the main model, which was pretrained using standard self-supervised learning methods on the same dataset. Here, assume that we have an ideal pretrained main encoder of which each output component corresponds to the latent factor of data variation (Zimmermann et al., 2021). While this ideal encoder should seamlessly adapt to downstream classification tasks, if most downstream task samples are bias-aligned, they may misguide the model to upweight spuriously correlated latent factors, leading to a biased solution despite well-generalized representations. We refer to this problem as the biased downstream application (Related analysis in Appendix C.1).

Table 3: (Supervised learning) Bias-conflict and unbiased accuracy (%) on MIMIC-CXR + NIH. Each ✓marker represents whether the model requires information on dataset bias. Bias ratio=10%.

| Accuracy | LNL ✓ | EnD ✓ | LfF ✗ | JTT ✗ | CVaR DRO ✗ | ERM ✗ | SimCLR ✗ | **Ours** ✗ |
|---|---|---|---|---|---|---|---|---|
| Conflict | $43.8_{\pm0.5}$ | $50.4_{\pm2.3}$ | $25.2_{\pm2.1}$ | $47.9_{\pm0.2}$ | $44.6_{\pm0.5}$ | $41.7_{\pm1.2}$ | $35.5_{\pm1.3}$ | $\mathbf{56.8}_{\pm1.7}$ |
| Unbiased | $68.1_{\pm1.0}$ | $\mathbf{71.8}_{\pm1.4}$ | $60.8_{\pm0.2}$ | $68.9_{\pm1.0}$ | $65.8_{\pm1.2}$ | $67.8_{\pm1.0}$ | $62.0_{\pm1.4}$ | $69.8_{\pm0.2}$ |

The above contradiction elucidates the importance of bias-conflicting samples, which serve as counterexamples of spuriously correlated feature components, thereby preventing the alleged involvement of such components in prediction. Based on these intuitions, we introduce a novel debiasing protocol that probes and upweights bias-conflicting samples to find and fully exploit feature components independent of spurious correlations. We apply our framework on two scenarios: linear evaluation and semi-supervised learning.

**Linear evaluation.** To validate our hypothesis on the biased downstream application, we conduct linear evaluation (Zhang et al., 2016; Oord et al., 2018) following the conventional protocol of self-supervised learning. Specifically, a linear classifier is trained on top of unsupervised pretrained representations by using target labels of training samples. After training a linear classifier $f_{W_b}^{cls}$ with pretrained biased encoder $f_\theta^{bias}$ given the whole training set $D = \{(x_k, y_k)\}_{k=1}^N$ with size $N$, an error set $E$ of misclassified samples and corresponding labels is regarded as bias-conflicting pairs. Then we train a linear classifier $f_{W_m}^{cls}$ on intentionally freezed representations of main encoder $f_\phi^{main}$ by upweighting the identified samples in $E$ with $\lambda_{up} > 0$. The loss function for *debiased* linear evaluation is defined as follows:

$$\ell_{debias}(D; W_m) = \lambda_{up} \sum_{(x,y) \in E} \ell(x, y; W_m) + \sum_{(x,y) \in D \setminus E} \ell(x, y; W_m),$$

where we use cross entropy loss for $\ell : \mathcal{X} \times \mathcal{Y} \times \mathcal{W} \to \mathbb{R}^+$. Note that the target labels are only used in training linear classifiers after pretraining.

Note that the debiased linear evaluation is not meant to compete directly with other supervised baselines. Instead, it aims to: (**a**) examine the potential origin of the failure in OOD generalization, (**b**) provide a rough estimate of the potential improvement achievable with frozen latent representations, and (**c**) compare with standard self-supervised baselines and identify the optimal learning algorithms, e.g. SimCLR (Chen et al., 2020), for training the main encoder.

**Semi-supervised learning.** We further compare our method directly to other supervised debiasing methods in the context of semi-supervised learning. Here we assume that the training dataset includes only a small amount of labeled data combined with a large amount of unlabeled data. As in linear evaluation, we train a linear classifier on top of the biased encoder by using labeled samples. After obtaining an error set $E$ of misclassified samples, we finetuned the whole main model by upweighting the identified samples in $E$ with $\lambda_{up}$. Note that supervised baselines are restricted to using only a small fraction of labeled samples, while the proposed approach benefits from the abundant unlabeled samples during pre-training of the biased encoder (Pseudo-code in the Appendix section A).

## 4 RESULTS

### 4.1 METHODS

**Dataset.** We evaluate several supervised and self-supervised baselines on **MIMIC-CXR + NIH** (Li et al., 2023), **UTKFace** (Zhang et al., 2017) and **CelebA** (Liu et al., 2015) in which prior work reported poor generalization performance due to spurious correlations (Dataset details in Appendix).

For MIMIC-CXR + NIH, we mixed the MIMIC-CXR (Johnson et al., 2019) and NIH (Wang et al., 2017) following Li et al. (2023) where the target categories are `no finding` and `pneumonia`. Most `pneumonia` images are collected from MIMIC-CXR, while most `no finding` images are from NIH. In other words, the biases come from systematic differences in data sources, where the classifier may erroneously rely on spurious radiographic features tied to variations in data acquisition pipelines (DeGrave et al., 2021) instead of true pathological indicators (Example images in Figure 8).

Table 4: (Linear evaluation) Bias-conflict and unbiased test accuracy (%) evaluated on UTKFace and CelebA. Models requiring information on target class or dataset bias in the (pre)training stage are denoted with ✓in columns Y and B, respectively. Our results are marked in bold to highlight the improvements compared to the mainly interested self-supervised learning baselines (Gray rows).

| Model | Y | B | UTKFace (age) | | UTKFace (gender) | | CelebA (makeup) | |
|---|---|---|---|---|---|---|---|---|
| | | | Conflict | Unbiased | Conflict | Unbiased | Conflict | Unbiased |
| LNL | ✓ | ✓ | $45.8_{\pm0.6}$ | $72.6_{\pm0.3}$ | $73.1_{\pm1.6}$ | $84.9_{\pm0.8}$ | $55.9_{\pm2.1}$ | $76.0_{\pm0.6}$ |
| EnD | ✓ | ✓ | $45.3_{\pm0.9}$ | $72.2_{\pm0.2}$ | $75.5_{\pm1.1}$ | $85.5_{\pm0.4}$ | $57.3_{\pm2.4}$ | $76.4_{\pm1.4}$ |
| JTT | ✓ | ✗ | $63.8_{\pm0.9}$ | $69.4_{\pm1.3}$ | $71.2_{\pm0.3}$ | $77.6_{\pm0.4}$ | $62.4_{\pm1.2}$ | $74.7_{\pm0.8}$ |
| CVaR DRO | ✓ | ✗ | $45.7_{\pm2.0}$ | $71.4_{\pm0.3}$ | $68.6_{\pm1.0}$ | $81.0_{\pm0.8}$ | $58.0_{\pm1.7}$ | $76.5_{\pm0.6}$ |
| ERM | ✓ | ✗ | $45.4_{\pm2.1}$ | $71.0_{\pm1.2}$ | $65.7_{\pm1.4}$ | $79.5_{\pm0.6}$ | $54.2_{\pm0.2}$ | $74.1_{\pm1.4}$ |
| SimSiam | ✗ | ✗ | $28.2_{\pm0.9}$ | $62.6_{\pm0.7}$ | $48.5_{\pm1.0}$ | $69.8_{\pm0.7}$ | $39.9_{\pm0.6}$ | $66.7_{\pm0.6}$ |
| VICReg | ✗ | ✗ | $32.3_{\pm0.6}$ | $64.6_{\pm0.3}$ | $51.0_{\pm1.4}$ | $71.3_{\pm0.7}$ | $48.6_{\pm0.6}$ | $71.9_{\pm0.2}$ |
| SimCLR | ✗ | ✗ | $36.4_{\pm1.5}$ | $66.3_{\pm0.6}$ | $56.3_{\pm0.2}$ | $74.2_{\pm0.2}$ | $46.9_{\pm1.0}$ | $69.8_{\pm0.4}$ |
| **DeFund** | ✗ | ✗ | $\mathbf{59.5}_{\pm0.8}$ | $\mathbf{70.6}_{\pm0.8}$ | $\mathbf{63.7}_{\pm2.0}$ | $\mathbf{74.9}_{\pm0.9}$ | $\mathbf{58.4}_{\pm0.6}$ | $\mathbf{73.1}_{\pm1.0}$ |

Table 5: (Semi-supervised learning) Accuracy results (%) on CelebA. Label fraction= 10%.

| Accuracy | CelebA (Makeup) | | | | | | | CelebA (Blonde) | |
|---|---|---|---|---|---|---|---|---|---|
| | LNL | EnD | JTT | CVaR DRO | ERM | SimCLR | **DeFund** | JTT | **DeFund** |
| Conflict | $55.7_{\pm1.4}$ | $55.3_{\pm1.5}$ | $51.5_{\pm1.9}$ | $55.6_{\pm1.5}$ | $51.5_{\pm1.1}$ | $50.5_{\pm4.7}$ | $\mathbf{60.5}_{\pm0.4}$ | $70.6_{\pm1.0}$ | $\mathbf{75.1}_{\pm0.8}$ |
| Unbiased | $75.6_{\pm0.5}$ | $\mathbf{76.2}_{\pm0.8}$ | $71.4_{\pm1.3}$ | $75.7_{\pm1.0}$ | $73.1_{\pm0.3}$ | $71.6_{\pm1.9}$ | $75.6_{\pm0.2}$ | $78.8_{\pm1.7}$ | $\mathbf{85.8}_{\pm0.3}$ |

For UTKFace, we conduct binary classifications using (`Gender`, `Age`) and (`Race`, `Gender`) as (target, spurious) attribute pair, which we refer to UTKFace (age) and UTKFace (gender), respectively. For CelebA, we consider (`HeavyMakeup`, `Male`) and (`Blonde Hair`, `Male`) as (target, spurious) attribute pairs, which are referred to CelebA (makeup) and CelebA (blonde), respectively. The results of CelebA (blonde) are reported in Appendix C.4. Following Nam et al. (2020); Hong & Yang (2021), we report bias-conflict accuracy together with unbiased accuracy, which is evaluated on the explicitly constructed validation set. We exclude the dataset in Figure 3 based on the observations that the SimCLR models are already invariant w.r.t spurious attributes.

**Baselines.** We mainly target baselines consisting of recent advanced self-supervised learning methods, SimCLR (Chen et al., 2020), VICReg (Bardes et al., 2021), and SimSiam (Chen & He, 2021), which can be categorized into contrastive (SimCLR) and non-contrastive (VICReg, SimSiam) methods. We further report the performance of vanilla networks trained by ERM, and other supervised debiasing methods such as LNL (Kim et al., 2019), EnD (Tartaglione et al., 2021), and upweighting-based algorithms, JTT (Liu et al., 2021) and CVaR DRO (Levy et al., 2020), which can be categorized into methods that leverage annotations on dataset bias (LNL, EnD) or not (JTT, CVaR DRO).

**Optimization setting.** Both bias and main encoder is pretrained with SimCLR (Chen et al., 2020) for 100 epochs on UTKFace, and 20 epochs on CelebA, respectively, using ResNet-18, Adam optimizer and cosine annealing learning rate scheduling (Loshchilov & Hutter, 2016). We use a MLP with one hidden layer for projection networks as in SimCLR. All the other baseline results are reproduced by tuning the hyperparameters and optimization settings using the same backbone architecture. We report the results of the model with the highest bias-conflicting test accuracy over those with improved unbiased test accuracy compared to the corresponding baseline algorithms, i.e., SimCLR for ours (More experimental details in Appendix D).

## 4.2 EVALUATION RESULTS

**Supervised learning.** To quantify the effectiveness of the rank regularization in-depth, we first consider a standard supervised debiasing scenario as similarly done in Table 2. For a MIMIC-CXR + NIH dataset, we found that the proposed framework outperforms other supervised baselines with

respect to bias-conflict accuracy. Table 14 in the Appendix shows that the rank-regularized networks effectively discover the bias-conflicting samples which are consistent with Table 1a, 1b, and 2.

**Linear evaluation.** We also found that DeFund outperforms every self-supervised baseline by a large margin in a linear evaluation protocol, including SimCLR, SimSiam and VICReg, with respect to both bias-conflict and unbiased accuracy (Table 4). Moreover, in some cases, DeFund even outperforms ERM models or supervised debiasing approaches regarding bias-conflict accuracy. Note that there is an inherent gap between ERM models and self-supervised baselines, roughly $8.7\%$ on average. Moreover, we found that non-contrastive learning methods generally perform worse than the contrastive learning method. This warns us against training the main model using a non-contrastive learning approach, while it may be a viable option for the biased model. Results of the proposed framework with non-contrastive learning methods are provided in the Appendix section C.5.

**Semi-supervised learning.** To compare the performance of supervised and self-supervised methods in a more practical and fair scenario, we sample $10\%$ of the labeled CelebA training dataset at random for each run. The remaining $90\%$ samples are treated as unlabeled ones and engaged only in pretraining encoders for self-supervised baselines. Labeled samples are provided equally to both supervised and self-supervised methods.

Remarkably, Table 5 and Table 16 in Appendix show that the proposed framework outperforms other state-of-the-art supervised debiasing methods. Existing upweighting protocols, such as JTT, fail to prevent deep networks from memorizing minority counterexamples. However, the proposed framework can fully utilize unlabeled samples with contrastive learning to prevent memorization. Existing bias-conflicting sample mining algorithms may be affected by the implicit bias of overparameterized networks, but this is unlikely to happen with the proposed framework since it only trains a simple linear classifier on top of a frozen biased encoder to identify such samples.

| Method | UTKFace (age) | | UTKFace (gender) | | CelebA (makeup) | |
|---|---|---|---|---|---|---|
| | Conflict | Unbiased | Conflict | Unbiased | Conflict | Unbiased |
| SimCLR | 36.4 | 66.3 | 56.3 | 74.2 | 46.9 | 69.8 |
| + Rank reg | 26.6 | 61.3 | 50.9 | 70.3 | 43.9 | 68.3 |
| + Upweight | 53.0 | 64.6 | 58.3 | 74.5 | 50.1 | 70.4 |
| **DeFund** | **59.5** | **70.6** | **63.7** | **74.9** | **58.4** | **73.1** |

(a) Ablation study

| Method | UTKFace (age) | | UTKFace (gender) | | CelebA (makeup) | |
|---|---|---|---|---|---|---|
| | Precision | Recall | Precision | Recall | Precision | Recall |
| SimCLR | 68.31 | 44.63 | **33.36** | 39.59 | 52.25 | 28.23 |
| **DeFund** | **68.67** | **75.94** | 29.98 | **50.93** | **55.29** | **32.46** |

(b) Precision and recall

Table 6: (**a**) Ablation study on introduced modules. (**b**) Precision and recall ($\%$) of bias-conflicting samples identified by SimCLR and our biased model. Both case used linear evaluation.

**Ablation study.** To quantify the extent of performance improvement achieved by each introduced module, we compared the linear evaluation results of (**a**) vanilla SimCLR, (**b**) SimCLR with rank regularization, (**c**) SimCLR with upweighting error set $E$ of the main model, and (**d**) DeFund. Note that (**c**) does not use a biased model at all. Table 6a shows that every module plays an important role in OOD generalization. Considering that the main model is already biased to some extent, we found that bias-conflict accuracy can be improved even without a biased model, where the error set $E$ of the biased model further boosts the generalization performance. We also measures the precision and recall of identified bias-conflicting samples in E, finding that the biased model detects more diverse bias-conflicting samples than the baseline (Table 6b). The improvement of recall in CelebA may seem marginal, but it is significant given the larger number of samples compared to UTKFace.

**Computational costs.** The proposed framework is computationally affordable as it only trains the linaer classifier (linear eval.) or finetune networks with a few epochs, e.g., about 30 epochs for UTKFace in debiasing stage. Self-supervised pre-training and linear evaluation takes 19.3 and 4.5 minutes with a single NVIDIA GeForce RTX 2080Ti, respectively.

## 5 CONCLUSION

**Contributions.** We present a novel solution to the challenging self-supervised debiasing, an important problem that has received little attention so far. Specifically, we (**a**) unveil the inductive bias towards encoding low effective rank representations in the presence of spurious correlations. Based on these findings, we (**b**) design a rank regularization that amplifies the feature redundancy by reducing the spectral entropy of latent representations. Then we (**c**) design a debiasing framework empowered by the biased model pretrained with abundant unlabeled samples.

## 6   ETHICS AND REPRODUCIBILITY STATEMENTS

**Ethics statement.** While this work has focused on encoding biased representations, more advances should be made in learning both biased and debiased representations. We found that explicit decorrelation of feature components in SimCLR does not lead to debiased representations. Moreover, one potential negative impact of the proposed framework could be the perpetuation of biases in data that are already present in society. The self-supervised biased encoder may amplify existing biases in the data, which could lead to further discrimination of certain populations. This must be prevented through proper regulation.

**Reproducibility statements.** We upload a file containing the code for our main experiments as a supplementary material. Furthermore, to ensure maximum reproducibility, pseudo-codes and hyper-parameter configurations are described in the Appendix A and D, respectively. Rest assured, we will fully open-source our code and pretrained models to reproduce all experiments in the paper.

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

# Appendix

The supplementary material is organized as follows. We begin with providing the algorithm of DeFund, followed by more discussions on the related works. Then we provide additional results and analyses in section C. Optimization setting, hyperparameter configuration and other experimental details are provided in section D.

## A  PSEUDOCODE

We provide the pseudo-code of the supervised and self-supervised version of the proposed debiasing framework here due to the limited space. The only minor difference between these two versions simply lies in the choice of training methodology, specifically whether the main and biased models are trained using supervised or self-supervised learning algorithms, respectively. Table 2 and 3 are based on Algorithm 2, while Table 4, 5, 6a and 6b are based on Algorithm 1.

---

**Algorithm 1** Debiasing Framework with unlabeled data (DeFund, self-supervised learning)

---

1: **Input:** $D_l = \{(x_k, y_k)\}_{k=1}^{N_1}$, $D_u = \{x_k\}_{k=1}^{N_2}$ for semi-supervised learning ($N_2 \gg N_1$), or $\varnothing$ for linear evaluation, $D = D_l \cup D_u$, batch size $n$, structure of $f^{bias}$ and $f^{main}$.

2:

3: **Stage 1.** *Pretraining encoders*

4: **for** subsampled minibatch $X = \{x_k\}_{k=1}^n$ from D **do**

5:     Update $\theta$ of $f_\theta^{bias}$ with SimCLR NT-Xent loss and $\lambda_{reg}\ell_{reg}(X; \theta)$.

6:     Update $\phi$ of $f_\phi^{main}$ with SimCLR NT-Xent loss.

7: **end for**

8: Obtain pretrained parameters $\hat{\theta}$ and $\hat{\phi}$.

9:

10: **Stage 2.** *Downstream task*

11: Freeze $f_{\hat{\theta}}^{bias}$ and train $f_{W_b}^{cls}$ with $D_l$. Identify the error set $E \subset D_l$ with trained $f^{bias}$.

12: **if** Linear evaluation **then**

13:     Freeze $f_{\hat{\phi}}^{main}$ and train $f_{W_m}^{cls}$ with $\ell_{debias}(D_l; W_m)$

14: **else if** Semi-supervised learning **then**

15:     Finetune $f^{main}$ with $\ell_{debias}(D_l; W_m, \phi)$ where $\phi$ is initialized with $\hat{\phi}$.

16: **end if**

---

**Algorithm 2** Debiasing Framework with rank regularization (DeRank, supervised learning)

---

1: **Input:** $D = \{(x_k, y_k)\}_{k=1}^N$, batch size $n$, structure of $f^{bias}$ and $f^{main}$.

2:

3: **Stage 1.** *Training biased model*

4: **for** subsampled minibatch $X = \{x_k\}_{k=1}^n, Y = \{y_k\}_{k=1}^n$ from D **do**

5:     Update $\theta, W_b$ of $f^{bias}$ with standard cross entropy loss and $\lambda_{reg}\ell_{reg}(X; \theta)$.

6: **end for**

7: Obtain pretrained parameters $\hat{\theta}$ and $\hat{W}_b$.

8: Identify the error set $E \subset D$ with trained $f^{bias}$.

9:

10: **Stage 2.** *Training main model*

11: Train $f^{main}$ with $\ell_{debias}(D; W_m, \phi)$.

---

## B  MORE RELATED WORKS

**Learning debiased representations.** Robinson et al. (2021) proposes an opposite-directional approach compared to our framework to improve generalizations of self-supervised learning. It aims to overcome the feature suppression and learn a wide variety of features by Implicit Feature Modification (IFM), which adversarially perturbs feature components of the current representations used to discriminate instances, thereby encouraging the encoder to use other informative features.

We observed that IFM improves the bias-conflict accuracy by about 1% on UTKFace (age) in Table 7, which is roughly consistent with the performance gains on the standard benchmarks, e.g., STL10, reported in the original paper. However, its performance gain is relatively marginal compared to the proposed framework.

Table 7: Results of Implicit Feature Modification (Robinson et al., 2021) with SimCLR on UTKFace (age). we denote $\epsilon$ as the adversarial budget of feature modification as in the original paper.

| Accuracy | SimCLR | $\epsilon = 0.05$ | $\epsilon = 0.1$ | $\epsilon = 0.5$ |
|---|---|---|---|---|
| Bias-conflict (%) | 36.4 | **37.5** | 36.4 | 33.7 |
| Unbiased (%) | 66.3 | **66.5** | 66.2 | 64.6 |

**Discovering bias without supervision.** In practice, several limitations exist against gleaning more labeled samples: labeling budget, expert-level knowledge required for labeling, data privacy, etc. In this regard, most training samples lack annotations on the spuriously correlated attributes.

To mitigate these problems, several works aim to discover biases without bias annotations. Liu et al. (2021) reveals that the standard ERM model may serve as a bias-capturing model if one trains it with strong capacity control. Yaghoobzadeh et al. (2019) shows that forgettables, or examples that have been forgotten at least once, contain more minority examples, and proposes a novel robust learning framework by fully exploiting the identified forgettable examples. Li & Xu (2021) obtains a biased attribute hyperplane of the generative models, which can help identify semantic biases by generating bias-traversal images. Li et al. (2022) introduces the discoverer model, which uncovers multiple unknown biases such that the difference of averaged predicted probabilities on the target attribute in two groups is maximized. Lang et al. (2021) proposes a novel framework, StylEx, which trains a styleGAN to specifically visualize multiple attributes underlying the classifier decisions.

While substantial advances have been made in discovering the unknown biases of neural networks without bias labels, these works still inevitably require target labels. In contrast, we consider a very challenging scenario that has received little attention so far: self-supervised debiasing. In this regard, our work addresses the following open problems/questions:

- Can we learn biased/debiased representations by using unlabeled samples?
- What is the fundamental difference between biased and debiased representations?
- Is supervised debiasing robust despite decreasing the number of labeled samples?
- How can bias-conflicting samples be discovered by leveraging information from unlabeled samples?
- Many recent works have reported the limitations of self-supervised learning (SSL) in OOD generalization. How can we overcome such limitations?

**Mitigating bias with reweighting.** Recently, Kirichenko et al. (2022) have reported an intriguing observation: Simple last layer retraining, so-called Deep Feature Reweighting (DFR), can match or outperform state-of-the-art approaches on spurious correlation benchmarks. Kirichenko et al. (2022) shows that biased classifiers still often learn core features associated with the desired attributes of the data. Based on these observations, they probe invariant features for the reweighting by leveraging explicit group-balanced dataset $\hat{D}$.

We compare the proposed framework with DFR as follows. First, while DFR and the proposed framework can mitigate the bias in representations by retraining the last linear layer, our method is not restricted to such last-layer retraining. Instead, the semi-supervised learning scenario is a more practical application of the proposed method. Specifically, we can fine-tune representations by fully exploiting both unlabeled and labeled samples, which improves the performance compared to the last layer retraining in Table 5. In contrast, DFR trains a linear classifier while freezing the pretrained representations as-is. More importantly, DFR requires pretrained networks or fully labeled datasets where we consider a more challenging scenario without such assumptions. Moreover, DFR does not use mining bias-conflicting samples in the training set. Specifically, DFR trains a new classification head from scratch on the available group-balanced data $\hat{D}$. In Kirichenko et al.

(2022), the reweighting dataset $\hat{D}$ often consists of a random group-balanced subset of the training or validation data. In other words, DFR is not designed to identify the bias-conflicting samples but exploits the existing group annotations. Considering practical situation with several limitations against collecting more labeled samples, it remains unclear how to obtain the group-balanced dataset $\hat{D}$ with sufficient number of samples in the absence of prior information on the dataset bias. In contrast, the proposed framework can leverage the explicit set $\hat{D}$ if accessible, *as well as* identifying the unknown bias-conflicting samples in the training set.

## C    ADDITIONAL RESULTS

Our additional results can be roughly categorized into: (1) more observations related to the rank reduction, (2) rank regularization in self-supervised learning, and (3) an examination of the potential of existing hyperparameters as a bias controller. Our observations include the rank reduction trends in CIFAR-10C and Vision Transformer (ViT, Dosovitskiy et al. (2020)), followed by rank regularization results with a moderate level of bias, and results of nuclear norm regularization. Then we present a simple synthetic simulation on the behavior of rank-regularized encoder. Then the potential of using shallow networks as the bias-capturing model will be discussed, followed by additional results on non-contrastive methods, MIMIC-CXR + NIH, and CelebA (blonde). Lastly, we provide additional analysis on relations between existing hyperparameters of self-supervised learning and effective rank.

### C.1    MORE OBSERVATIONS

**Rank reduction.** Figure 5a shows that the rank of latent representations from a penultimate layer of classifier decreases as the bias ratio increases in CIFAR-10C. In Table 8, we supplement the unbiased test accuracy of CMNIST and CIFAR-10C from the experiments presented in Figure 2c and 5a, respectively. Moreover, similar rank reduction trends are observed in Vision Transformer (ViT, Dosovitskiy et al. (2020)). We train ViT on CMNIST and CIFAR-10C for 2000 and 10000 iterations, respectively, with Adam optimizer of learning rate 0.001, patch size 4, dimension of output tensor 128, number of transformer blocks 6, number of heads in multi-head Attention layer 4, dropout rate 0.2 and dimension of the MLP (FeedForward) layer 1024. Figure 5b, 5c show that the effective rank of the output of the Transformer encoder $\mathbf{z}_L^0$ (notation follows the original paper) decreases as bias ratio increases.

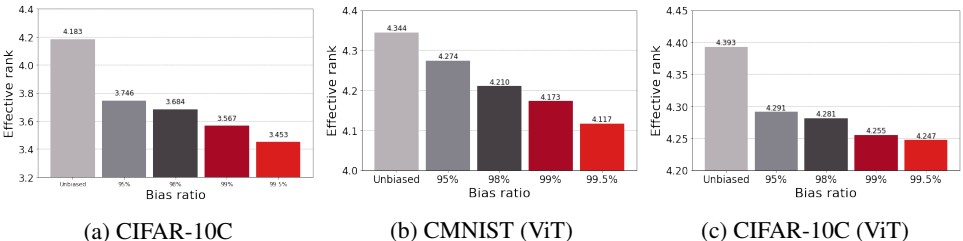

| (a) CIFAR-10C | (b) CMNIST (ViT) | (c) CIFAR-10C (ViT) |

Figure 5: Effective rank measured with (**a**) CIFAR-10C (ResNet-18), (**b**) CMNIST (ViT) and (**c**) CIFAR-10C (ViT).

Table 8: Unbiased test accuracy (%) on CMNIST and CIFAR-10C measured with varying bias ratio $r$. The model trained with unbiased dataset ($r = 10\%$) serves as a baseline.

| Dataset | Unbiased | $r = 95\%$ | $r = 98\%$ | $r = 99\%$ | $r = 99.5\%$ |
|---------|----------|------------|------------|------------|--------------|
| CMNIST | 99.87 | 88.27 | 68.13 | 36.21 | 13.61 |
| CIFAR-10C | 78.71 | 46.15 | 34.18 | 26.76 | 20.94 |

**Rank regularization with moderate level of bias.** To study the compatibility of rank regularization with weak spurious correlations, we apply the rank regularization to the moderately biased CMNIST, i.e., bias ratio=60%. Table 9 shows that the rank regularization works well in this natural setting. This implies that the rank regularization can be leveraged to reveal the moderate level of bias embedded in the representations, which is supported by the empirical results of other general datasets, e.g., Waterbirds, UTKFace or CelebA.

Table 9: Ablation study of rank regularization on weakly biased CMNIST (Bias ratio=60%). Our rank-regularized model is trained with $\lambda_{reg} = 50$. For a fair comparison, all the other experimental settings are fixed. Bias-aligned accuracy, bias-conflict accuracy, precision and recall of identified bias-conflicting samples are reported.

| Methods | Align (%) | Conflict (%) | Precision (%) | Recall (%) |
|---------|-----------|--------------|---------------|------------|
| ERM | 99.49 | 97.81 | 79.55 | 0.87 |
| Ours | 96.25 | 38.15 | 91.56 | 60.97 |

**Nuclear norm regularization.** While the proposed rank regularization controls the auto-correlation matrix inspired from 2a, one may regularize nuclear norm of the latent representations, which is a convex relaxation of a matrix rank.

To compare the quality of biased representations, we call DeFund$_{nu}$ as the proposed debiasing framework with normalized nuclear norm regularization, instead of Eq. (3). Specifically, for a normalized nuclear norm, the absolute singular values are summed and then divided with the feature dimension. From our preliminary analysis in Table 10 below, the performance of nuclear norm regularization was underperformed by the proposed rank regularization in Eq. (3). Moreover, for the case of the nuclear norm, top singular values are significantly large, as shown in Figure 2d, so that the distributional property of singular values may be obfuscated in the nuclear norm as shown in Table 11. This suggests that while nuclear norm may be a candidate for rank regularizer with a solid theoretical background, we recommend using the effective rank in feature analysis.

Table 10: (Linear evaluation) Bias-conflict and unbiased test accuracy (%) evaluated on UTKFace and CelebA. DeFund$_{nu}$ refers to the proposed framework with nuclear norm regularization.

| Model | UTKFace (age) | | CelebA (makeup) | |
|-------|---------------|----------|-----------------|----------|
| | Conflict | Unbiased | Conflict | Unbiased |
| DeFund$_{nu}$ | $53.9_{\pm 0.3}$ | $67.5_{\pm 0.3}$ | $52.1_{\pm 0.5}$ | $72.5_{\pm 0.1}$ |
| **DeFund** | $\mathbf{59.5}_{\pm 0.8}$ | $\mathbf{70.6}_{\pm 0.8}$ | $\mathbf{58.4}_{\pm 0.6}$ | $\mathbf{73.1}_{\pm 1.0}$ |

Table 11: Normalized nuclear norm (norm / dimension) measured in CMNIST and CIFAR-10C with varying bias ratios.

| Dataset | Unbiased | 95(%) | 98(%) | 99(%) | 99.5(%) |
|---------|----------|-------|-------|-------|---------|
| CMNIST | 2.47 | 2.56 | 2.56 | 2.59 | 2.46 |
| CIFAR-10C | 7.12 | 5.92 | 6.34 | 6.54 | 6.51 |

**Behavior of rank-regularized encoder.** Here, we present a simple simulation which conceptually clarifies the impacts of rank regularization in self-supervised learning. Inspired from Chen et al. (2020); Robinson et al. (2021), we create a DigitsOnSTL10 dataset as in Figure 6a where MNIST images are randomly selected and placed on top of the STL10 images. After self-supervised representation learning, we train two independent linear classifiers on top of the freezed representations, where we provide label of foreground MNIST digit for one classifier, and label of background STL10 object class for the other. After training linear classifiers, we measure the ratio of MNIST classifier test accuracy to STL10 classifier test accuracy, which we treat as a proxy of ratio of spuriously correlated features to invariant features, i.e., degree of bias in representations. Intuitively, the proposed bias metric increases as the encoder focus more on the short-cut attribute, i.e., MNIST digit.

We measure the bias metric on the representations of ResNet-18 encoders trained by SimCLR (Chen et al., 2020) together with rank regularization loss $\lambda_{reg}\ell_{reg}$, where $\lambda_{reg} > 0$ is a balancing hyperparameter. As denoted in the main paper, we apply regularization not on the output of projection networks but directly on the output of base encoder, which makes it fully agnostic to networks architecture. Figure 6b shows that the rank regularization exacerbates the "feature suppression"

phenomenon revealed by Chen et al. (2021). The representation becomes more biased as it is trained with stronger regularization. While the overall performance of self-supervised learning may be upper-bounded due to the constraint on effective dimensionality (Jing et al., 2021), we observe in Figure 6b that the bias-conflict accuracy is primarily sacrificed compared to the bias-aligned accuracy. Coupled with results in section 4, this result implies that rank regularization can amplify bias in self-supervised encoder.

Moreover, we have conducted an additional experiment to better understand the biased downstream application problem. We first train the encoder on unbiased CMNIST using SimCLR. By unbiased, we mean that the background color in the training images is randomly assigned, unlike the images shown in Figure 4d. Subsequently, we train the linear classifier on top of the encoder using (**a**) Biased CMNIST samples with a bias ratio of 99.5%, and (**b**) Unbiased CMNIST samples.

As shown in Table 12, training the linear classifier with unbiased samples (**b** case) leads to the unbiased model, which works evenly well on every group. Despite training the encoder on a fully unbiased dataset, the use of biased samples in the downstream task results in a significant drop in the bias-conflict test accuracy. These findings highlight the potential risks associated with using biased training samples directly in downstream applications, as biased samples may inadvertently involve spurious factors that are correlated with the bias (such as the background color in this example).

Table 12: Linear evaluation results on the CMNIST with varying bias ratio in the downstream dataset.

| Bias ratio | 99.5% | Unbiased (10%) |
|---|---|---|
| Aligned (%) | 99.79 | 96.27 |
| Conflict (%) | 69.99 | 96.14 |

Table 13: Comparison study on the depth of biased networks. Both networks are trained with target labels on CIFAR-10C (Bias ratio=95%). For UTKFace (age) and CelebA (makeup), both networks are pretrained with SimCLR followed by last linear layer training. Reported in (%).

| Networks | CIFAR-10C | | UTKFace (age) | | CelebA (makeup) | |
|---|---|---|---|---|---|---|
| | Precision | Recall | Precision | Recall | Precision | Recall |
| Shallow | 64.73 | **59.50** | 55.68 | 69.98 | 27.49 | **33.79** |
| ResNet-18 | **71.39** | 51.43 | **68.67** | **75.94** | **55.29** | 32.46 |

## C.2 SHALLOW NETWORK

Considering the inductive bias of neural networks towards encoding low effective rank representations in this paper, one may ask whether the shallow neural networks can easily learn such simple inductive

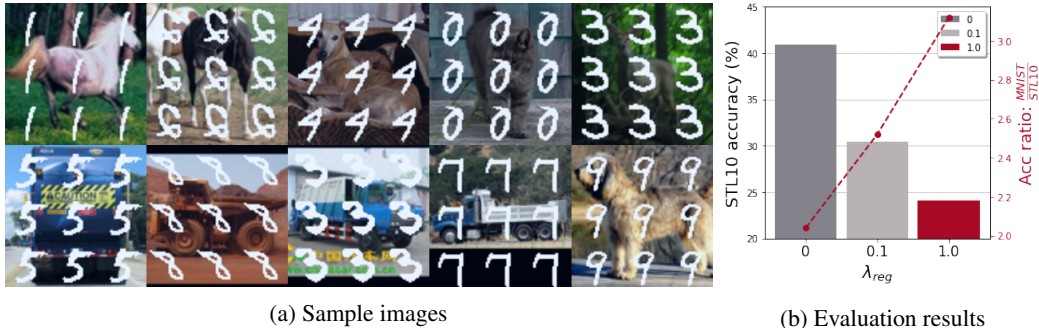

(a) Sample images                    (b) Evaluation results

Figure 6: (**a**) Sample images from DigitsOnSTL10 dataset. (**b**) Test accuracy of STL10 classifier and bias metric.

Table 14: (MIMIC-CXR + NIH) (**a**), (**b**): Precision and recall of identified bias-conflicting samples. (**c**), (**d**): bias-aligned and bias-conflicting accuracy (%) of ERM and our rank-regularized model.

|  | (**a**) Precision ($\uparrow$) | (**b**) Recall ($\uparrow$) | (**c**) BA ($\uparrow$) | (**d**) BC ($\downarrow$) |
|---|---|---|---|---|
| ERM | 52.21 | 54.31 | 95.15 | 29.75 |
| Rank reg. | **55.93** | **70.37** | **96.55** | **17.10** |

bias and serve as a bias-capturing network. In this regard, we observe some pros and cons of using a shallow network as the bias model throughout experiments. Specifically, we use a simple convolutional network with three convolution layers as a counterpart of ResNet-18, with feature map dimensions of 64, 128 and 256, each followed by a ReLU activation and a batch normalization.

In the labeled setting, CIFAR-10C in Table 13 shows a tradeoff between precision and recall of the shallow network: The shallow network improves the recall of identified hard samples, i.e., the fraction of the bias-conflicting samples that are identified, because it is robust to the unintended memorization due to their fewer number of hyperparameters. However, it sacrifices the precision, i.e., the fraction of identified samples that are indeed bias-conflicting because its performance on the bias-aligned samples is degraded due to the low expressivity.

While the shallow network shows promising results with a simple dataset, the tradeoff worsens in the self-supervised setting with a larger dataset. Table 13 shows that the shallow network may suffer from bad precision. It is conventional wisdom that unsupervised learning benefits more from bigger models than its supervised counterpart (Chen et al., 2020). Considering this, the general performance of shallow networks may deteriorate in a large-scale self-supervised learning scenario. In this case, the identified error set $E$ contains too many false-positive bias-conflicting samples. While one may improve the performance with good care of hyperparameter tuning, e.g., depth of networks, learning rate, etc., it may be more laborious compared to the proposed framework, which has only a few scalar hyperparameters, e.g., $\lambda_{reg}$.

### C.3 Additional results on MIMIC-CXR + NIH

For a MIMIC-CXR + NIH dataset, we report precision and recall of identified bias-conflicting proxies in Table 14, showing that the proposed rank-regularization improves the minority mining performance.

### C.4 Additional results on CelebA

We report the results of CelebA (blonde) in here due to the limited space. Detailed information on the dataset and simulation settings is provided in the section D. Following Sagawa et al. (2019); Liu et al. (2021), we report worst-group and average accuracy because CelebA (blonde) includes abundant samples in (`Blonde Hair`=0, `Male`=0) bias-conflicting group. The number of training samples in each group is provided in Table 21.

Table 15 shows that DeFund outperforms not only every self-supervised baseline, but also ERM, CVaR DRO, and LfF Nam et al. (2020) in linear evaluation. Table 16 shows that DeFund outperforms all the other baseline methods in semi-supervised learning, which is consistent with Table 5 of the main paper.

Moreover, recent works unveil that CelebA (blonde) exhibits a large class imbalance which in turn correlates with a large group imbalance. Recent studies Hong & Yang (2021); Idrissi et al. (2022) found that both target classes are biased toward a non-Male bias class in CelebA (blonde) which obfuscates whether the dataset is indeed biased. In this regard, Idrissi et al. (2022) observed that the simple class balancing serves as a powerful baseline due to the class imbalance. This directly motivates us to alleviate the class imbalance and focus on the dataset bias itself. Following Hong & Yang (2021), we randomly subsample images from (`Blonde Hair`=0, `Male`=0) group so that two target classes are biased toward different bias classes. The number of training samples before and after subsampling is provided in Table 21d and 17b, respectively. Table 17a shows that DeFund outperforms JTT with respect to both worst-group and average accuracy, where its bias-conflict-

accuracy-version is also reported in Table 5 of the main paper. These additional results imply that the proposed framework ensures reliable performance in the presence of strong spurious correlations.

Table 15: (Linear evaluation) Worst-group and average accuracy (%) evaluated on CelebA (blonde). Results of ERM, CVaR DRO, LfF (Nam et al., 2020) and JTT are come from Table 1 of the original JTT paper (Liu et al., 2021). Each first and second ✓marker represents whether the model requires information on target class or dataset bias in pretraining stage, respectively.

| Accuracy | ERM ✓✗ | CVaR DRO ✓✗ | LfF ✓✗ | JTT ✓✗ | VICReg ✗✗ | SimSiam ✗✗ | SimCLR ✗✗ | **DeFund** ✗✗ |
|---|---|---|---|---|---|---|---|---|
| Worst-group | 47.2 | 64.4 | 77.2 | **81.1** | 10.2 | 1.1 | 17.1 | **77.9** |
| Average | **95.6** | 82.5 | 85.1 | 88.0 | 89.0 | 89.0 | 88.9 | **89.0** |

Table 16: (Semi-supervised learning) Worst-group and average accuracy evaluated on CelebA (blonde). Label fraction is set to 10%. Each first and second ✓marker represents whether the model requires information on target class or dataset bias in pretraining stage, respectively.

| Accuracy | LNL ✓✓ | EnD ✓✓ | JTT ✓✗ | CVaR DRO ✓✗ | ERM ✓✗ | SimCLR ✗✗ | **DeFund** ✗✗ |
|---|---|---|---|---|---|---|---|
| Worst-group (%) | 40.3 | 41.5 | 79.2 | 49.1 | 30.8 | 12.8 | **80.8** |
| Average (%) | **91.1** | 91.0 | 91.0 | 91.0 | 89.1 | 89.1 | 90.0 |

| Methods | Worst-group (%) | Average (%) |
|---|---|---|
| JTT | 70.6 | 86.6 |
| **DeFund** | **75.1** | **94.8** |

(a) Accuracy

| | | Male | |
|---|---|---|---|
| | | 0 | 1 |
| Blonde | 0 | 1558 | 53483 |
| | 1 | 18417 | 1102 |

(b) Subsampled CelebA (blonde)

Table 17: (Semi-supervised learning) (**a**) Worst-group and average accuracy evaluated on subsampled CelebA (blonde). Label fraction is set to 10%. (**b**) Number of training samples for each group in subsampled CelebA (blonde). (Original dataset in Table 21d)

## C.5 NON-CONTRASTIVE METHODS

We provide the results of proposed framework implemented based on non-contrastive methods. Specifically, we leverage SimSiam (Chen & He, 2021) and VICReg (Bardes et al., 2021) as baselines. Table 18 shows that the generalization performance of both baselines can be improved with the proposed debiasing framework.

## C.6 HYPERPARAMETER ANALYSIS

While rank regularization biases the representations effectively, we do not argue that it is the optimal form of semantic bottleneck but rather that it highlights the unrecognized importance of controlling effective rank in encoding biased representations. In this regard, we examine the impacts of existing optimization hyperparameters on the effective rank and degree of bias in latent representations. Specifically, we investigated four candidates of bias controller through the lens of effective rank and generalizations: hardness concentration parameter $\beta$ of hard negative sampling (Robinson et al., 2020), temperature $\tau$ in InfoNCE (Oord et al., 2018) loss, strength of $\ell_2$ regularization $\lambda_{\ell_2}$ and the number of training epochs $T$.

**Hardness concentration parameter.** Recent works (Robinson et al., 2020; Cai et al., 2020; Tabassum et al., 2022) stress out the importance of negative examples that are difficult to distinguish from an anchor point. Several recent works propose algorithms on selecting informative negative

|            | Conflict | Unbiased |
|------------|----------|----------|
| SimSiam    | 28.15    | 62.63    |
| + Rank reg | 23.40    | 59.65    |
| + Upweight | 56.12    | 65.44    |
| **DeFund**$_\text{Siam}$ | **60.37** | **67.78** |

(a) SimSiam

|            | Conflict | Unbiased |
|------------|----------|----------|
| VICReg     | 32.33    | 64.58    |
| + Rank reg | 29.73    | 62.08    |
| + Upweight | 51.19    | 63.41    |
| **DeFund**$_\text{VIC}$ | **53.93** | **66.31** |

(b) VICReg

Table 18: Bias-conflict accuracy and unbiased accuracy evaluated on UTKFace (age). Last row corresponds to the full version of proposed framework which upweights misclassified samples identified by biased model. Results are averaged on 4 different random seeds. Accuracy is reported in (%).

samples, often controlled by hardness concentration parameter $\beta$ (Robinson et al., 2020) coupled with importance sampling. Robinson et al. Robinson et al. (2021) conducted a synthetic simulation showing that increasing $\beta$ makes instance discrimination tasks more difficult, thereby enforcing the encoder to represent more complex features. Thus we aim to examine whether $\beta$ can contribute to learn a debiased representations with real-world dataset.

**Temperature.** A recent work on contrastive loss (Wang & Liu, 2021) have revealed that temperature $\tau$ can also control the strength of penalties on hard negative samples. Contrastive loss with high temperature turns out to be less sensitive to the hard negative samples (Robinson et al., 2020; 2021), thereby encouraging representations to be locally clustered while the uniformity of features on the hypersphere decreases (Wang & Isola, 2020). That being said, we hypothesized that the temperature $\tau$ may indirectly affect the effective dimensionality of representations, where large $\tau$ may decrease the effective rank.

$\ell_2$ **regularization and early-stopping.** Recent studies Sagawa et al. (2019; 2020) underline the importance of regularization for worst-case generalization where the naive upweighting strategy may fail if it is not coupled with strong regularization that prevents deep networks from memorizing upweighted bias-conflicting samples. In this regard, Liu et al. (2021) leverages capacity control techniques, e.g., strong $\ell_2$ regularization or early-stopping, to train complexity-constrained bias-capturing models. We investigate whether such regularizations can serve as a bias controller in self-supervised learning as well.

| Accuracy | 0.01 | 0.05 | 0.1 | 0.15 | 1 |
|----------|------|------|-----|------|-----|
| Conflict | 35.8 | 36.3 | 37.5 | 37.6 | 36.6 |
| Unbiased | 65.6 | 65.6 | 66.6 | 66.5 | 66.0 |

(a) Biased linear evaluation

|          | SimCLR | $\beta$=0.1 |
|----------|--------|-------------|
| Conflict | 62.0   | 64.2        |
| Unbiased | 78.9   | 80.7        |

(b) Debiased linear evaluation

Table 19: Results of controlling concentration parameter $\beta$ on UTKFace (age). Accuracy is reported in (%). (**a**): Accuracy of linear evaluation without upweighting bias-conflicting samples. Each value in top row indicates $\beta$ used in pretraining. (**b**) Accuracy of linear evaluation with upweighting ground-truth bias-conflicting samples. Both models use $\lambda_{up} = 10$.

**Results.** We evaluate each knob on generalizations with SimCLR. Table 20 and 19a show that impacts of both early-stopping and concentration parameter $\beta$ on generalizations are marginal, in contrast to the observations reported in supervised learning or synthetic simulations (Robinson et al., 2021). However, it still remains unclear whether the debiased representations can be encoded by controlling $\beta$. It is because the model may reach a biased solution even though it encodes debiased representations, if most samples in linear evaluation are bias-aligned, as discussed in the main paper. To preclude such confounding relationships, we conduct debiased linear evaluation with upweighting ground-truth bias-conflicting samples. Table 19a and 19b show that there was no significant difference in the performance gain of $\beta$ in biased and debiased linear evaluation, which implies that $\beta$ is not enough to fully debias representations.

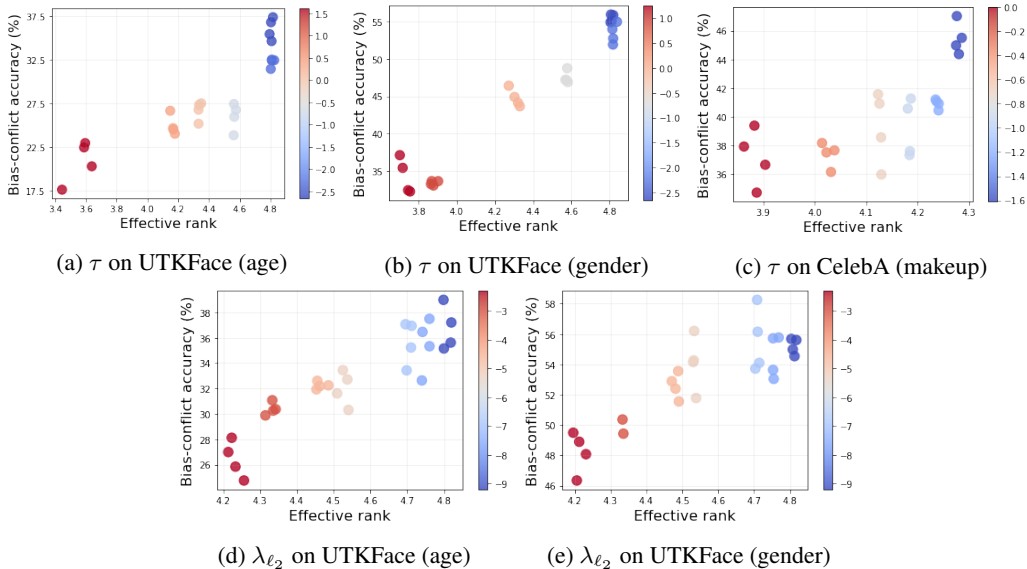

(a) $\tau$ on UTKFace (age)   (b) $\tau$ on UTKFace (gender)   (c) $\tau$ on CelebA (makeup)

(d) $\lambda_{\ell_2}$ on UTKFace (age)   (e) $\lambda_{\ell_2}$ on UTKFace (gender)

Figure 7: Analysis on temperature $\tau$ and strength of $\ell_2$ regularization $\lambda_{\ell_2}$. Effective rank and bias-conflict accuracy are measured with varying $\tau$ for (**a**, **b**, **c**), and $\lambda_{\ell_2}$ for (**d**, **e**). Standard deviation of bias-aligned accuracy on each experiment is 1.0%, 2.8%, 0.3%, 1.3% and 1.7% in order. Performance become quickly degenerated as $\lambda_{\ell_2}$ increases over 0.005 in CelebA (makeup).

Despite the failure of learning debiased representations with controlling $\beta$, biased representations can be learned by controlling temperature $\tau$, and strength of $\ell_2$ regularization in some cases. Figure 7a, 7b and 7c show that effective rank, temperature and bias-conflicting accuracy are highly correlated each other in both UTKFace and CelebA. It implies that the effective rank can serve as a metric of generalization performance and degree of bias in representations. While temperature control cannot be generalized to several non-contrastive learning methods (Chen & He, 2021; Bardes et al., 2021; Zbontar et al., 2021), this results imply that the temperature may serve as an effective bias controller for contrastive learning methods using InfoNCE loss. Moreover, stronger-than-typical $\ell_2$ regularization also limits the effective rank and bias-conflict accuracy to some extent in UTKFace (Figure 7d and 7e), while it fails to do so in CelebA.

This series of observations afford us a novel insight that both explicit (rank regularization) and implicit (temperature control, strong $\ell_2$ regularization) methods offer a way to train biased representations. However, it still remains unclear how to directly learn *debiased* representations. While increasing temperature or reducing effective rank bias representations, inverse does not always hold; Abnormally small temperatures cause the contrastive loss only focus on the nearest one or two samples, which heavily degenerates the performance (Wang & Liu, 2021). Moreover, we found that explicit decorrelation of feature components in SimCLR does not lead to debiased representations (not shown in figure).

To sum up, we provide useful recipes on learning biased representations, where rank regularization is mainly discussed in the main paper due to its intuitive insights, good performance and broad applicability. We hope these discussions facilitate in-depth studies about advanced algorithms on learning both biased and debiased representations in unsupervised manner.

## D   EXPERIMENTAL SETUP

### D.1   DATASETS

We mainly evaluate our debiasing framework on MultiCMNIST (Li et al., 2022), MIMIC-CXR + NIH (Li et al., 2023), UTKFace (Zhang et al., 2017) and CelebA (Liu et al., 2015) in which several prior works has observed poor generalization performance due to spurious correlations. Example images are presented in Figure 8.

Table 20: Results of early-stopping on UTKFace. We denote $T$ as the number of training epochs.

| Attribute | Accuracy | $T = 5$ | $T = 10$ | $T = 15$ | $T = 20$ | $T = 25$ |
|---|---|---|---|---|---|---|
| Age | Bias-conflict (%) | 31.6 | 33.0 | 32.4 | 32.8 | 32.8 |
| | Unbiased (%) | 63.3 | 64.1 | 63.6 | 63.7 | 63.7 |
| Gender | Bias-conflict (%) | 54.6 | 54.0 | 53.5 | 53.4 | 54.5 |
| | Unbiased (%) | 72.1 | 72.0 | 71.8 | 72.2 | 72.7 |

**MultiCMNIST.** It is worth noting that existing off-the-shelf synthetic biased datasets often fail to account for real-world scenarios in which multiple bias attributes can coexist simultaneously. To address this limitation, the work by Li et al. (2022) introduces the innovative Multi-Color MNIST (MultiCMNIST) dataset, designed to emulate complex real-world multi-bias scenarios. Specifically, there are two bias attributes, namely `left color` and `right color`, where we set bias ratio=99% for the left color and bias ratio=95% for the right color.

**MIMIC-CXR + NIH.** The dataset discussed here serves as a poignant example of spurious correlations within medical imaging datasets. In such datasets, machine learning classifiers may struggle to discern the true underlying pathological indicators, such as the presence of pneumonia, often relying on spurious radiographic features tied to variations in data acquisition procedures (DeGrave et al., 2021). To simulate spurious correlations in medical imaging dataset, we mix MIMIC-CXR (Johnson et al., 2019) and NIH (Wang et al., 2017) datasets into a MIMIC-CXR + NIH dataset following Li et al. (2023). The original NIH contains 50500 `no finding` and 876 `pneumonia` training images, while the original MIMIC-CXR has 10145 `no finding` and 7209 `pneumonia` training images. Given the scarcity of `pneumonia` images in the NIH dataset, we curate the MIMIC-CXR + NIH dataset by primarily extracting `pneumonia` images from MIMIC-CXR and `no finding` images from NIH. In MIMIC-CXR + NIH, the target categories are `no finding` and `pneumonia`, and the biases come from two data sources. It contains 8000 training images with a bias ratio of 0.9, 250 unbiased validation images, and 250 unbiased test images.

**UTKFace.** We first consider UTKFace dataset which is consist of human face images with varying `Race, Gender` and `Age` attributes. For each sensitive attribute, we categorize all samples into two groups. Specifically, for label associated with age, we assign 1 to samples with age $\leq 10$, and 0 to samples with age $\geq 20$ following (Hong & Yang, 2021). For label associated with race, we assign 1 to samples with race $\neq$ white, e.g., Black, Indian and Asian, and 0 to samples with race $=$ white. For label associated with gender, we assign 1 to female, and 0 to male. Based on this settings, we conduct binary classifications using (`Gender, Age`) and (`Race, Gender`) as (target, spurious) attribute pairs. Following Hong & Yang (2021), we construct a biased dataset by randomly truncating a portion of samples, where roughly 90% of samples are bias-aligned in our setting. Pixel resolutions and batch size are $64 \times 64$ and 256, respectively.

**CelebA.** For CelebA, we consider (`HeavyMakeup, Male`) and (`Blonde Hair, Male`) as (target, spurious) attribute pairs, following Nam et al. (2020); Hong & Yang (2021); Sagawa et al. (2019). Pixel resolutions and batch size are $256 \times 256$ and 128, respectively. The exact number of samples for each prediction task is summarized in Table 21.

| | | A | | | | G | | | | M | | | | M | | | | Data | |
|---|---|---|---|---|---|---|---|---|---|---|---|---|---|---|---|---|---|---|---|
| | | 0 | 1 | | | 0 | 1 | | | 0 | 1 | | | 0 | 1 | | | NIH | MIMIC |
| G | 0 | 8229 | 822 | R | 0 | 4354 | 534 | H | 0 | 25789 | 54460 | B | 0 | 57214 | 53483 | P | 0 | 3600 | 400 |
| | 1 | 134 | 1346 | | 1 | 435 | 5344 | | 1 | 49804 | 163 | | 1 | 18417 | 1102 | | 1 | 400 | 3600 |
| (a) UTKFace (A) | | | | (b) UTKFace (G) | | | | (c) CelebA (H) | | | | (d) CelebA (B) | | | | (e) MIMIC+NIH | | | |

Table 21: Number of training samples for each prediction task. `A` for `Age`, `G` for `Gender`, `R` for `Race`, `M` for `Male`, `H` for `HeavyMakeup`, `B` for `Blonde Hair`, and `P` for `Pneumonia`.

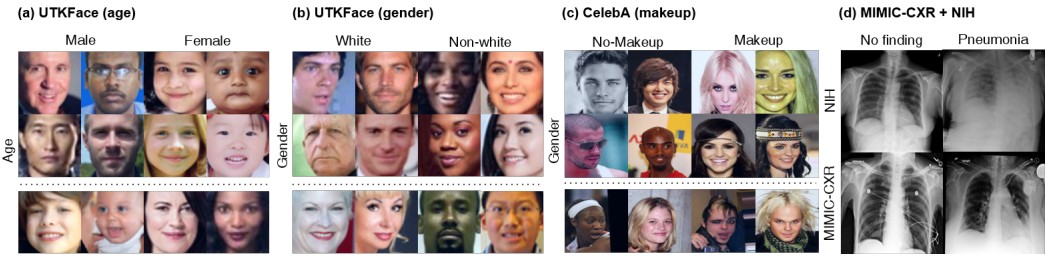

Figure 8: Example images of datasets. Top-row annotations refer to the target attributes, i.e. `Gender`, `Race`, `HeavyMakeup` and `Pneumonia`, while the left-side annotations refer to the bias attributes, i.e. `Age`, `Gender` and data source, respectively. For (**a**), (**b**), and (**c**), the images above the dotted line denote the bias-aligned samples, while the ones below the dotted line are the bias-conflicting samples.

## D.2 RANK REDUCTION & REGULARIZATION ANALYSIS

**CMNIST & MultiCMNIST.** For CMNIST, we use a simple convolutional network with three convolution layers as a counterpart of ResNet-18, with feature map dimensions of 64, 128, and 256, each followed by a ReLU activation and a batch normalization. The convolutional network is trained for 2000 iterations using SGD optimizer with initial learning rate 0.1 and decaying by 0.1 for every 600 iterations, following Zhang et al. (2021). For a MultiCMNIST, the experimental settings including neural architecture and optimizer follow the original paper (Li et al., 2022) for a fair comparison.

**CIFAR10-C and Waterbirds.** For CIFAR10-C and Waterbirds in Figure 3c, we use ResNet-18 and ResNet-50 with pretrained weights provided in PyTorch torchvision implementations, respectively. ResNet-18 is trained for 10000 iterations using the Adam optimizer with learning rate 0.001. After training, misclassified training samples are identified as the bias-conflicting samples as in Table 1a. Following the official implementation of JTT, ResNet-50 is trained for 300 epochs, and early-stopped with referring to the validation accuracy, using SGD optimizer with learning rate 0.0001.

**Hyperparameters.** In Table 1a, $\lambda_{reg} = 35$ and $\lambda_{reg} = 20$ are used for CMNIST and CIFAR-10C, respectively. In Table 1b, $\lambda_{reg} = 10$ is used.

## D.3 DEBIASING EXPERIMENTS

**Architecture details.** We use ResNet-18 back-bone architecture with pretrained weights provided in in PyTorch torchvision implementations. For projection networks in SimCLR, we use the MLP consists of one hidden layer with feature dimension of 512, followed by a ReLU activation. We employ a single linear classifier in downstream tasks for all self-supervised learning methods.

**Training details.** For MIMIC-CXR+NIH, both biased and main classifiers are trained by using Adam optimizer with learning rate of 0.0003. Biased and main classifiers are trained for 5 and 100 epochs, respectively. For a rank regularization, $\lambda_{reg} = 10$ is used. For a upweighting, $\lambda_{up} = 5$ is used with $\lambda_{\ell_2} = 0.0005$.

Both biased and main encoders are pretrained for 100 epochs on UTKFace, and 20 epochs on CelebA, by using Adam optimizer with learning rate of 0.0003. Cosine annealing scheduling (Loshchilov & Hutter, 2016) is leveraged with warmup for the first 20 epochs on UTKFace, and 4 epochs for CelebA.

For biased encoders, we apply rank regularization with using $\lambda_{reg}$ of 0.3, 0.5, 0.01 and 0.03 for UTKFace (age), UTKFace (gender), CelebA (makeup) and CelebA (blonde), respectively. This values are selected by tuning $\lambda_{reg} \in \{0.0, 0.1, 0.3, 0.5, 1.0\}$ for UTKFace and $\lambda_{reg} \in \{0.0, 0.01, 0.02, 0.03, 0.05\}$ for CelebA. Specifically, we report the results of model with highest worst-group accuracy (for CelebA (blonde)), or bias-conflicting test accuracy over those with improved unbiased test accuracy compared to the SimCLR baseline. Same values are consistently used for upweighting in ablation study (Table 6a). To emphasize the contribution of rank regulariza-

tion, we do not control any other parameters, e.g., strength of $\ell_2$ regularization, temperature $\tau$, or number of training epochs. Specifically, we fix $\tau = 0.07$ and $\lambda_{\ell_2} = 0.0001$ for every experiment.

After pretraining, we conduct either linear evaluation or finetuning with using $\lambda_{up}$ of 10, 5, 8 and 15 for UTKFace (age), UTKFace (gender), CelebA (makeup) and CelebA (blonde), respectively. For UTKFace and CelebA (makeup), these values are selected by tuning $\lambda_{up} \in \{5, 8, 10\}$ using the above-mentioned decision rules, where $\lambda_{up} \in \{5, 8, 10, 15\}$ is compared for CelebA (blonde). Same values are consistently used in ablation study (Table 6a). For linear evaluation, we train a linear classifier on top of pretrained main encoder for 3000 iterations on UTKFace, and 5000 iterations on CelebA, with using learning rate of 0.0003 and upweighting identified bias-conflicting samples. For semi-supervised learning, we finetune the whole main model for 5000 iterations, with using SGD optimizer, momentum of 0.9, $\lambda_{\ell_2} = 0.1$, learning rate of 0.0001, and $\lambda_{up} = 8, 15$ for CelebA (makeup) and CelebA (blonde), respectively.

**Data augmentations.** Following SimCLR, we generate multiviewed batch with random augmentations of (a) random resized crop with setting the scale from 0.2 to 1, (b) random horizontal flip with the probability of 0.5, (c) random color jitter (change in brightness, contrast, and saturation) with the probability of 0.8 and scale of 0.4, (d) random gray scaling with the probability of 0.2. In linear evaluation and finetuning, we only apply random horizontal flip. Same augmentation pipeline is applied to both SimSiam and VICReg.

**Baselines.** For a fair comparison, we tune hyperparameters of other baselines using the same ResNet-18 back-bone architecture. We use the official implementation of JTT which also includes that of CVaR DRO. Other baselines are reproduced by ourselves with referring to original papers. LNL is trained for 20 epochs on UTKFace, and 40 epochs on CelebA and MIMIC-CXR + NIH, with using Adam optimizer and learning rate of 0.001. For EnD, we set the multipliers $\alpha$ for disentangling and $\beta$ for entangling to 1. For JTT, we tune the upweighting factor $\lambda_{up} \in \{20, 50, 80\}$ and number of training epochs $T \in \{30, 40, 50\}$, following the original paper. For CVaR DRO, we tune the size of the worst-case subpopulation $\alpha \in \{0.1, 0.2, 0.5\}$. For SimSiam and VICReg, the architectures for the additional layers followed the official implemenation of each method, where the hyperparameters for the training is identical to the SimCLR case. For C.5, $\lambda_{reg} = 0.001$ for **DeFund**$_{\text{Siam}}$ and $\lambda_{reg} = 0.1$ for **DeFund**$_{\text{VIC}}$.

