# OpenReview forum: "Self-supervised debiasing using low rank regularization"
_ICLR.cc/2024/Conference — ICLR 2024 Conference Withdrawn Submission_

### Official Review · Reviewer_xxLf · 2023-10-29

**Soundness:** 2 fair
**Presentation:** 3 good
**Contribution:** 2 fair
**Rating:** 3
**Confidence:** 3

**Summary:**

This paper aims to investigate and reduce the spurious correlations inside the data and neural net and hence improve the generalization ability of the prediction model. In addition, the paper tries to tackle this problem in the setting of using unlabeled samples.  They empirically observe that the rank of the dataset representation matrix is related to the dataset bias. By this logic, they use rank regularization to define their own regularization term and conduct experiments to show the effectiveness of their debias framework.

**Strengths:**

1. The paper tries to tackle an important question regarding improving the generalization ability of models from the aspect of spurious correlation.
2. Empirically, they have some interesting findings about the effective rank of the representation matrix and the bias of the dataset
3. They also show their method has better performance under their experiment settings.
4. The use of unlabeled data to improve the method.

**Weaknesses:**

While the paper aims to tackle an important question of improving model generalization ability by reducing bias, many of the concepts are not well supported.

1. the bias is not rigorously defined, which makes the problem hard to connect to the ultimate goal of improving generalization.
2. The whole logic of the paper is built on providing a method that evaluates its own craft benchmark dataset, which makes the paper less credentialed.
2. all the statements come from empirical observation and lack of analysis support or more empirical support.
3. The paper does not experiment on the real dataset, i.e. whether their method actually has an overall improvement on the generalization ability on most of the benchmark datasets(the datasets without intentionally synthesizing bias inside)
4. The paper should also show in what kind of bias the performance will be good and also what kind of situation the method does not work.

**Questions:**

see weakness

---

### Official Review · Reviewer_bvf8 · 2023-10-31

**Soundness:** 3 good
**Presentation:** 3 good
**Contribution:** 2 fair
**Rating:** 6
**Confidence:** 4

**Summary:**

This work attempts to solve a classification problem in a biased dataset. To this end, the authors employ a self-supervised framework where they add another loss apart from the SimCLR NT-Xent loss which is referred to as rank regularization. The motivation behind rank regularization seems to be the fact that correlated latent space vectors account for more learning of the bias and hence poor performance on the bias conflicting group of samples. This loss makes the backbone image encoder aware of the bias that is present in the dataset. Once this model is trained, the samples on which this model classifies erroneously are considered to be the bias conflicting samples and are subsequently upweighted to achieve good accuracy on both bias conflicting and bias aligned samples. Evaluations have been done on UTK-Face, CelebA datasets.

**Strengths:**

- The paper is very well written and is easy to follow
- The idea behind rank regularization is very interesting and seems to be very effective
- The improvements that has been achieved over the compared baselines are also impressive

**Weaknesses:**

- The main weakness appears to be in the choice of the contrastive learning objective. The backbone has been trained in the same way as in the SimCLR paper, however other constrastive learning techniques have not been explored. The regularization loss seems to be a very general one and hence can be applied to other contrastive learning techniques such as Barlow Twins [1] and BYOL [2]. I would especially encourage the authors to show the effect of this loss on [1] as the regularization loss proposed in this paper seems to be inspired from the redundancy reduction term in [1]. More advanced baselines would help position the manuscript in a much better place

- The method should be evaluated with more backbone networks. Some results on rank reduction have been provided in the Appendix with a ViT network, but similar results on CelebA and UTKFace would be very interesting and a valuable addition to the manuscript


[1] Zbontar, Jure, et al. "Barlow twins: Self-supervised learning via redundancy reduction." International Conference on Machine Learning. PMLR, 2021.
[2] Grill, Jean-Bastien, et al. "Bootstrap your own latent-a new approach to self-supervised learning." Advances in neural information processing systems 33 (2020): 21271-21284.

**Questions:**

See the weaknesses section. Apart from the two points mentioned there, the following questions arise:

- From what I understand the encoders are run with the losses on the datasets that are used in the downstream task. Self-supervision generally benefit from large dataset sizes. Can you please mention the dataset size on which the pre-training task has been done? Are all the datasets combined for this?
- Can the pre-training and the downstream datasets be different? This might lead to a domain mismatch in my opinion but can be an interesting direction to investigate.

---

### Official Review · Reviewer_Txeo · 2023-11-01

**Soundness:** 2 fair
**Presentation:** 2 fair
**Contribution:** 3 good
**Rating:** 3
**Confidence:** 4

**Summary:**

The paper introduces the observation that spuriously correlated attributes make neural networks inductively biased towards encoding lower effective rank representations and proposes a two-stage debiasing method. In the first stage, a biased model is trained with rank regularization without both target labels and bias labels. In the second stage, the main model is trained by upweighting the bias-conflicting samples which are identified by biased model.

**Strengths:**

In situations where bias labels are not provided, many studies have focused on debiasing algorithms; however, most of these assume the availability of target labels. This research, by assuming a scarcity of target labeled data, adds the advantage of considering a more realistic scenario.

**Weaknesses:**

W1. Throughout the paper, the mixture of supervised learning settings and self-supervised learning settings makes it challenging to comprehend.
- Initially, the paper emphasizes operating within a self-supervised learning setting, indicated by the title, abstract, and the early sections of the introduction. However, as the introduction progresses and throughout the remainder of the paper, it becomes ambiguous when and under what circumstances the supervised learning setting is also considered. This transition makes it challenging to discern the specific situations or settings being addressed as the paper progresses.
- In section 2, is the analysis assumed to be under a supervised learning setting for training the encoder, or does it assume a self-supervised learning setting?

W2. There are important related works that are not cited. Please review the following papers and provide citations.
1. Learning debiased classifier with biased committee, NeurIPS 2022
	This paper improves the proposed method by adopting a self-supervised representation as the frozen backbone. Since self-supervised learning is dependent on class labels, it is less affected by the spurious correlations between classes and latent attributes.
2. Spread spurious attribute: improving worst-group accuracy with spurious attribute estimation, ICLR 2022
3. SelecMix: Debiased Learning by Contradicting-pair Sampling, NeurIPS 2022
4. Correct-N-Contrast: A Contrastive Approach for Improving Robustness to Spurious Correlations, ICML 2022
5. Environment Inference for Invariant Learning, ICML 2021
6. Unsupervised Learning of Debiased Representations with Pseudo-Attributes, CVPR 2022

W3. The experimental design lacks consistency.
- Is the proposed algorithm expected to vary based on different settings? Having a unified algorithm might appear more cohesive. If not, it may be preferable to compare the supervised and self-supervised settings using a consistent benchmark. The variability in the experiments conducted for 'supervised learning,' 'linear evaluation,' and 'semi-supervised learning' poses a challenge in determining the superiority of the proposed method. (Could the authors clarify the reason for using different benchmarks for each experiment?)
- It would be helpful to include the JTT results for CMNIST and CIFAR-10C in Table 1. Presenting these results could potentially allow for the consolidation of Tables 1-a and 1-b into a single table.

W4.Training the encoder through self-supervised learning might potentially require a significant amount of time to reach completion. Additionally, concerns arise regarding the overall increase in training time and resource requirements when separately training both the main encoder and the biased encoder. Could you clarify the expected additional time required compared to Empirical Risk Minimization (ERM)?

W5. Before reviewing the algorithm table in the appendix, understanding how the encoder and linear layer are trained in each setting was quite challenging. To enhance clarity, it would be helpful to provide detailed explanations in the main paper so that these training methods are well understood without referring to the appendix.
An example of a question I had before reviewing the appendix:
- In supervised learning (Table 3), is the main encoder trained with CE loss? This was not explained.
- Regarding Linear evaluation, it's mentioned that the main encoder, f_\theta^{main}, is frozen. How is f_\theta^{main} trained?
- A summarized method and sequence for training the biased encoder, main encoder, biased classifier, and main classifier would be beneficial.

W6.The author emphasizes the exclusion of target labels during encoder training, reserving their use solely for the training of the linear classifier. However, incorporating target labels in the linear classifier is deemed essential. According to Kirichenko et al., training only the final layer is adequate for neural network debiasing.
- All methods in Table 4 utilize target labels during training. Whether these labels are used in pretraining becomes less significant if they are employed in training the linear classifier for debiasing. As a result, column Y appears redundant.
- Could you explain why DeFund exhibits poorer performance than ERM in UTKFace (gender)?

W7. In table 2, in row (1), the right color seems to be misdescribed as 'BC'. Is it supposed to be 'BA'?

**Questions:**

Suggestions
S1. To ensure a fair comparison, it would be beneficial if the experiments in Figure 4 (a), (b), and (c) all used the same input. Evaluating the reconstructed results based on a consistent input and comparing the degree of bias would provide a more appropriate basis for a thorough comparison.